# Analysis of the Actual Contact Surface of Selected Aircraft Tires with the Airport Pavement as a Function of Pressure and Vertical Load

**Mariusz Wesołowski \***, **Krzysztof Blacha, Paweł Pietruszewski and Paweł Iwanowski**

Air Force Institute of Technology, Airfield Division, ul. Ks. Bolesława 6, 01-494 Warsaw, Poland;
krzysztof.blacha@itwl.pl (K.B.); pawel.pietruszewski@itwl.pl (P.P.); pawel.iwanowski@itwl.pl (P.I.)
*   Correspondence: mariusz.wesolowski@itwl.pl; Tel.: +48-261-851-324

**Abstract:** The contact surface of the wheel with the airport surface is important for the safety of flight operations in the ground manoeuvring area. The area of the contact surface, its shape and stress distribution at the pavement surface are the subject of many scientists' considerations. However, there are only a few research studies which include pressure and vertical load directly and its influence on tire-pavement contact area. There are no research studies which take into account aircraft tires. This work is a piece of an extensive research project which aims to develop a device and a method for continuous measurement of the natural airport pavement's load capacity. One of the work elements was to estimate the relationship between wheel pressure and wheel pressure on the surface, and the area of the contact surface. The results of the research are presented in this article. Global experience in this field is cited at the beginning of the article. Then, the theoretical basis for calculating the wheel with the road surface contact area was presented. Next, the author's research views and measurement method are presented. Finally, the obtained test results and comments are shown. The tests were carried out for four types of tires. Two of them were airplane tires from the PZL M28 Skytruck/Bryza and Sukhoi Su-22 aircraft. Two more came from the airport ASFT (airport surface friction tester) friction tester-one smooth ASTM; the other smooth retreaded type T520. The tires were tested in a pressure range from 200 to 800 kPa. The range of vertical wheel load on the pavement was 3.23–25.93 kN for airplane tires, and 0.8–4.0 kN for friction tester tires. The tests proved a significant influence of the wheel pressure value and wheel pressure on the surface on the obtained contact surface area of the wheel with the surface. In addition, it was noted that the final shape and size of the contact surface is affected by factors other than wheel pressure, tire pressure and dimensions.

**Keywords:** airfield pavements; airplane tire; tire-surface contact area; soil; load-bearing capacity

## 1. Introduction

The contact zone between the wheel and the surface is important for air operations by aircraft in the ground manoeuvring area. This aspect is important for both artificial and natural surfaces. By contacting the tire with the artificial surface, aircraft forces are transferred to the ground. The correct transfer guarantees the safety of flight operations, and thus the safety of crew and passengers. In the case of natural surfaces, the correct impact of the wheel on the surface increases safety in situations of emergency artificial runway excursion or safe flight operations on the airport's natural (grass) functional elements.

Research on pavement response to wheel load has been conducted for many years. Already, in 1989 [1], the impact of tire pressure and its type on the response of a flexible surface has been analysed. They included three types of tires in their tests: radial, diagonal and broad-profile radial,

and performed tests for three pressure levels. They proved that each of the factors presented has a significant impact on the formation of stress and deformation in the pavement.

The tests of the wheel-surface contact surface were conducted by [2]. Researchers took a standard 11R22-5 truck tire. The research was carried out with the Heavy Vehicle Simulator (HVS) Mark VI. The finite element method (FEM) was used for the analysis of the research results. The analysis results confirmed that, for a given pressure, the contact surface decreased with the increase of the tire pressure. Similar tests were performed [3]. They tested two tires for trucks. The tests were carried out using five different values of vertical load in the range from 26.6 kN to 79.9 kN, and four pressure values in the wheel in the range from 552 kPa to 862 kPa. The analysis was conducted in the context of pavement deteriorations resulting from incorrect tire pressure values and axle loads exceeding the permissible values.

In [4], the research aimed to correlate the tire-road contact surface with the parameter of the anti-skid properties of the pavement. In their work, researchers determined the actual contact surface of a tire fragment with three-dimensional pavement models, reflecting nearly 30 different pavements with surfaces of different texture depth values. The tests were carried out on a micro scale and did not include the pressure in the wheel. However, they relatively faithfully reflected the influence of the pavement texture parameter on its anti-skid properties. The researchers compared the obtained contact surface results with the actual friction coefficient results determined with the T2Go device. The author [5] also conducted tests of the wheel-road contact area in the context of driving safety. The work emphasizes the impact of improper wheel pressure on the deterioration of the vehicle's driving characteristics, the effectiveness of braking systems, traction control systems and decreased riding comfort. Although the work [5] does not say anything directly about the contact surface of the wheel with the road surface, it shows many negative aspects related to improper tire pressure, which affects the adhesion of the tire to the road surface, and thus deteriorating traction. In [6], attention was paid to the problem of incorrect tire pressure and the impact of the phenomenon on the results of braking and suspension diagnostics. In their research, it was noticed that, at low measuring speeds of 5–7 km/h, the tire pressure does not significantly affect the braking forces obtained on the rollers. Although, as in the case of [5], the wheel-surface contact surface was not examined, it was indicated that changes in pressure in the wheel significantly change its traction parameters. Moreover, [7] studied the impact of pressure and temperature on the obtained friction coefficient values. Both the finite element method and experimental experiments were used for the analysis. It was proved that as the temperature increased, the friction coefficient decreased. A similar situation occurred in the case of pressure, where the friction coefficient decreased with its increase.

In [8], the problem of the wheel-pavement contact surface was considered in a mechanistic way and the stress on the wheel-pavement contact was determined using the finite element method. The analysis was carried out in a static manner, which means that the stresses were determined under the wheel under equilibrium conditions. At the time of work publication [8], it was not possible to measure the forces under the wheel rolling at high speed due to their complicated spatial distribution. The Euler–Lagrangian model was used for the calculations. The stress distribution in three directions was determined, and the significant influence of load, tire pressure and anti-skid properties of the pavement on its parameters was proven. Work was carried out in the context of early diagnosis of pavement deteriorations. Works on the pavement response assessment to a typical, double or single truck tire load were conducted by [9]. In their research, they used three-dimensional stress distribution in a truck-loaded structure. They have proven that overloaded trucks with incorrect wheel pressure can cause significant damage to the road surface. The same authors also considered the use of stress at the wheel with the pavement contact in the design of the road pavement, based on its life expectancy [10]. Similar studies were conducted by [11], who in their work, presented an assessment of the impact of tire size and wheel pressure on the stress on the surface of the wheel-surface contact, whereas [12] examined the stress at the interface near the pavement under a moving wheel. The purpose of his work was to develop a method enabling this type of measurement. The stress distribution under the

wheel was also analysed in [13]. The research was conducted in the context of highway pavement damage by wide truck wheels. It has been shown that, in the shallow layers of the pavement, large spikes of stress caused by overloaded vehicles are possible.

An analysis of stress distribution in the subsoil using the finite element method was the subject of the work [14]. According to the tire, movement of the tire on the ground surface causes uneven distribution of stress in the ground and at the wheel-surface contact. In their work, they presented the influence of tire pressure, wheel load and soil moisture on soil propagation. They proved that the change in tire pressure and load caused a change in the shape of the vertical pressure distribution on the hard surface of dry soil, but these variables did not affect the distribution of vertical stress in soft wet soil or below a depth of 0.15 m. Similarly [15], the effect of anisotropic friction on tractive forces and lateral forces influencing the wheel-pavement contact using the finite element method was analysed. They carried out similar tests [16], which showed the relationships between the modulus of subgrade reaction, as a characteristic field parameter, and the load bearing capacity of the natural pavement. They conducted similar studies [17]. They focused on the effect of normal load on friction coefficient values at the contact surface of rubber and rough surface. They have developed a model that simply reflects the real contact surface and stress at the contact of two surfaces.

The contact surface between the wheel and the surface is also indirectly considered in agricultural sciences. Researchers in the article [18] presented a model of soil layer readiness for various methods of soil cultivation. According to the authors, the condition of the surface and the resulting stresses directly affect the complexity of carrying out some forms of tillage, which affects its costs. Some [19] checked the soil consolidation under a wheel with different values of the CC (compaction capacity) parameter and the load bearing capacity of such tires in different soil and water conditions. The research involved different soil types with different optimum density and porosity.

This article presents a piece of a more extensive project on the continuous testing of natural surfaces. The project was created due to the lack of exhaustive requirements for natural surfaces. International aviation documents [20–23] quite briefly treat natural airport pavements, which are highly important in the aspect of air traffic safety. The essence of the measurement is based on the phenomenon of airplane wheel depression in a natural airport pavement [24]. The wheel of the plane submitted to pressure on the surface creates a rut. This phenomenon is shown in Figure 1.

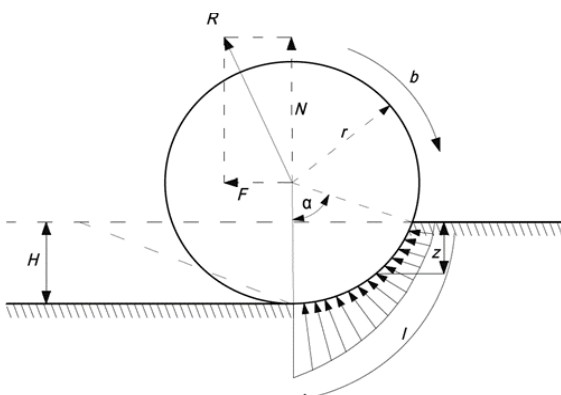

**Figure 1.** Forces distribution in a wheel rolling on a natural surface: N—vertical soil reaction to wheel load, F—friction force, R—resultant of F and N forces, H—rut depth, a—length of wheel and pavement contact section, $\alpha$—angle corresponding to section length a, r—wheel radius, b—rotation direction, z—position of the layer concerned [25].

The rut depth H from the above drawing can be determined from the Formula (1):

$$H = \frac{q_k^2 \cdot D}{\sigma^2 \cdot k_h} \tag{1}$$

where:

H—rut depth,
$q_k$—ground pressure of one landing gear wheel,
D—diameter of the aircraft wheel,
σ—soil strength,
$k_h$—coefficient depending on soil plasticity and tire stiffness.

In order to quickly determine the proper turf load bearing capacity, a maximum rut depth of 1/14 of the diameter of the main or nose wheel diameter can be assumed, as described in Formula (2).

$$H = \frac{D}{14} \tag{2}$$

The amount of estimated deformation proves that an aircraft moving on a natural pavement will be protected against damages resulting from ground motions that are too intense [25].

In order to clarify the measurement method, the authors developed a secondary method for determining the actual contact surface of the wheel with the road surface. Using different types of aircraft tires, a model illustrating the actual wheel with the surface contact surface was established, with different pressure and load conditions. This will allow the choice of appropriate input parameters for the natural surface assessment. The article presents differences between theoretical models and the results of laboratory tests.

After presenting the introduction containing a literature review in the subject area and an outline of the problem, the following sections will present the measurement method used, the measurement set built for the needs of the project, and the results of the research. Finally, the authors summarize the analysis and present the conclusions of the analysis and the directions for further work.

## 2. Materials and Methods

The method of calculating the wheel with the contact surface is not clear. Various formulae can be found in the literature, on the basis of which it is possible to determine the above value. Such formulae were presented in monographs by, among others [25] and [26]. A common and simple Formula is (3):

$$F = \frac{P}{p} \tag{3}$$

where:

F—wheel with the road contact area [$m^2$],
P—vertical load of the wheel on the road [N],
P—wheel pressure [Pa].

This is a simple formula that does not take into account many factors. However, it allows an approximate determination of the wheel contact surface with the road surface area with great accuracy and with undemanding calculations is a quick solution. The next formula cited in the literature is as follows (4):

$$F = (0.75 + 0.005 \times D) \times \frac{P}{p} \tag{4}$$

where:

F—wheel with the surface contact area [$m^2$],
P—vertical load of the wheel on the surface [N],
P—wheel pressure [Pa],
D—wheel diameter [cm].

This formula, apart from the parameters quoted in Formula (3), also takes into account the size of the wheel in the form of its diameter. The value (0.75 + 0.005 × D) is called the tire load factor and corrects the surface area by about 10%–12% [25]. This gives a closer view of the actual contact area. The formulae presented here do not involve the type of tire (radial or diagonal), the retreading method (smooth, smooth-retreaded or retreaded), as well as the shape of the tire. Tires with a contact area similar to a rectangle and those whose contact area is shaped as an ellipse are produced. In addition, rubber hardness and measurement temperature can also affect the final size of the contact area.

The authors, at the time of work related to a large research project, performed measurements of the actual contact area of the wheel with the road surface. For this purpose, two airplane tires from two types of aircraft were used—PZL M28 Skytruck/Bryza (Figure 2) and Su-22 (Figure 3).

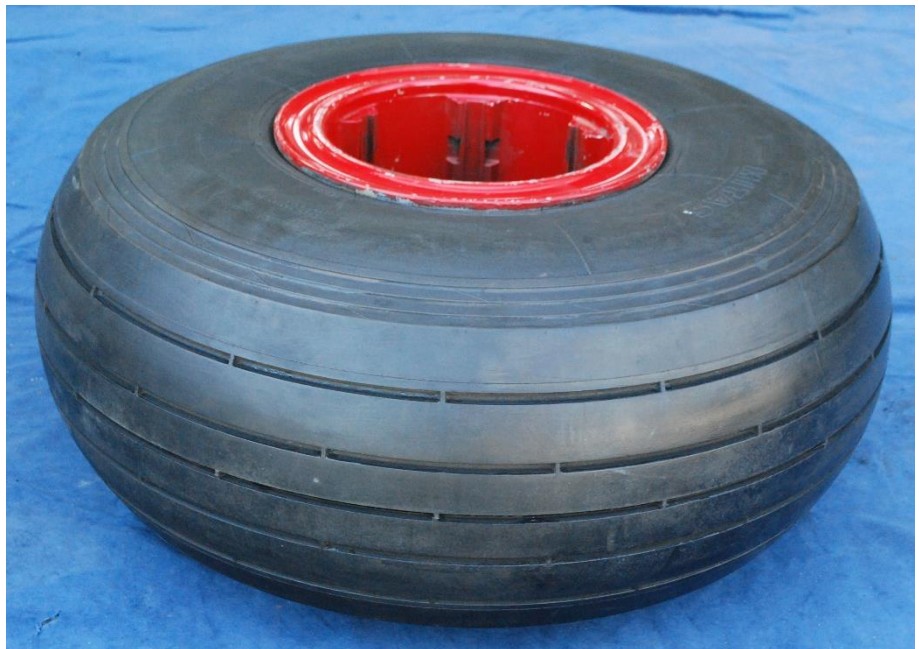

**Figure 2.** PZL M28 Skytruck/Bryza tire.

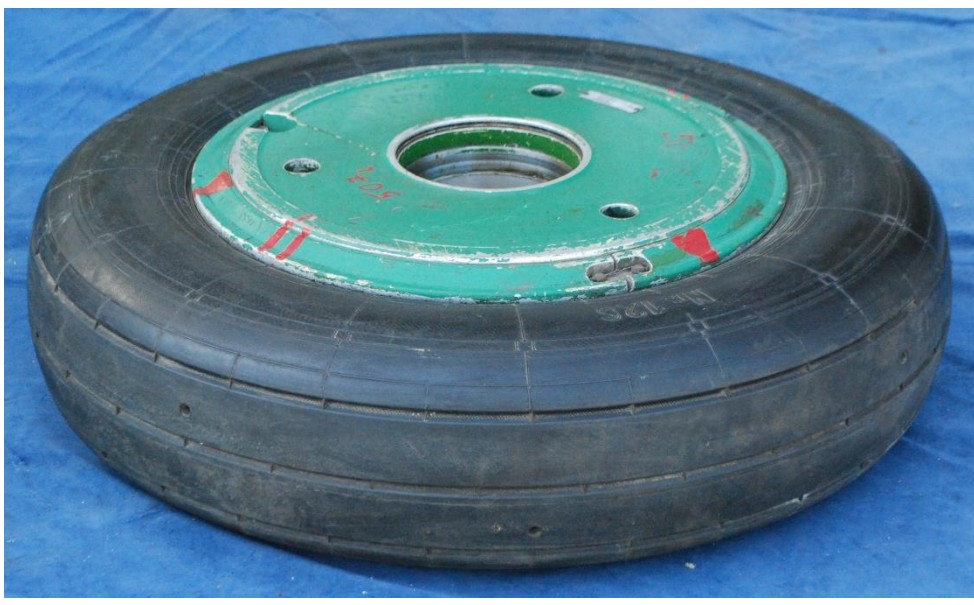

**Figure 3.** Su-22 tire.

PZL M28 Skytruck/Bryza is a turboprop transport aircraft. Its main tire is 310 mm wide and 720 mm high and is tubeless. Tire is filled with nitrogen at a standard up to 6 Bar pressure. Su-22 belongs to a group of jet military aircraft. Its main tire is 230 mm wide and 880 mm high. The tire is filled with nitrogen at a standard up to 16 Bar pressure. In addition, two ASFT (airport surface friction tester) anti-skid tires (Figures 4 and 5) were analysed. T520 tire is a smooth-tread tire, while ASTM E-1551 is a smooth tire. Both tires have the same dimensions with 4″ wide and 8″ rim diameter. Standard pressure during use is up to 7 Bar.

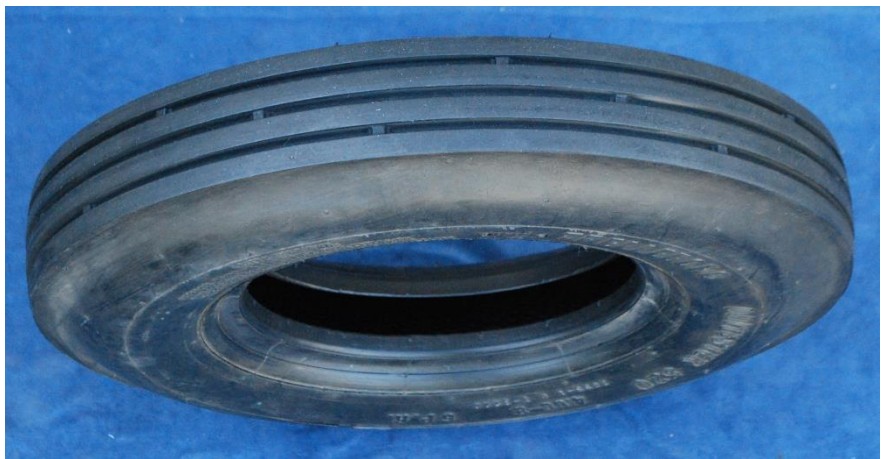

**Figure 4.** ASFT (airport surface friction tester) T520 tire.

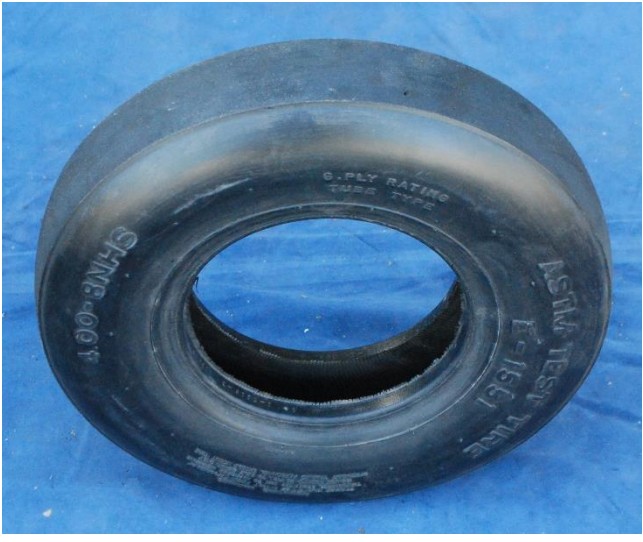

**Figure 5.** ASTM E-1551 tire.

*Procedure Characteristics*

Measurements were carried out in laboratory conditions, maintaining repeatability conditions. Temperature during tests was in the range 22 °C–22.5 °C. The test consisted of exerting a controlled pressure on a wheel, with a fixed pressure through a non-deformable plate. A contrast of waxes and lanolin with a black dye mixture was applied to the tire and an A3 paper screen was placed between the wheel and the plate. Each sheet was marked with the tire reference number, wheel pressure and pressure applied. While the plate is being pressed against the wheel, the contrast was marked on the screen, creating a trace of the contact area in the form of a negative. A proprietary test stand was built to perform the measurement. For this purpose, a non-deformable frame was made of thick-walled steel C-profiles, which is shown in Figure 6.

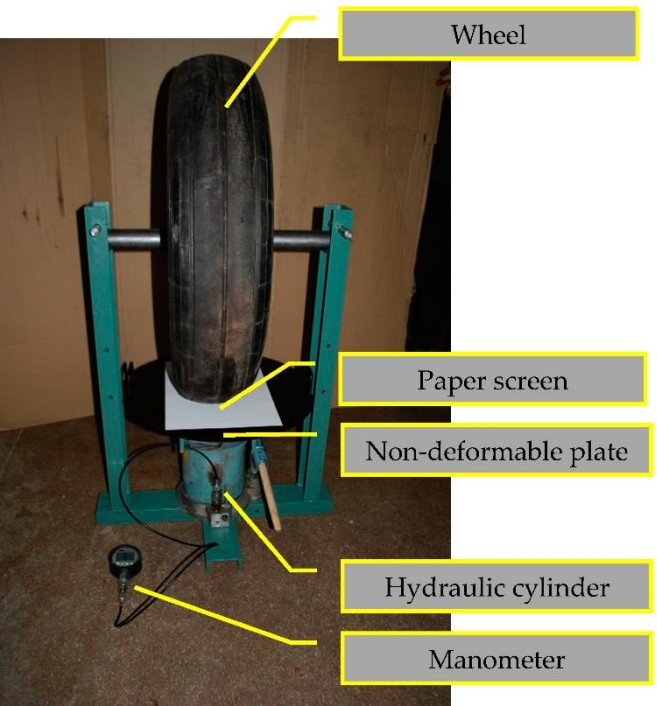

**Figure 6.** View of the test stand.

The frame has locks at the top for mounting the measuring wheel. The measuring wheel was screwed in specially prepared hubs, mounted on a steel rod with a diameter of 50 mm. A hydraulic cylinder with a digital manometer is located at the bottom of the frame. The manometer was previously calibrated in a testing machine, resulting in a correlation of the manometer indications (in MPa) to the force exerted by the actuator (in kN). A view of the correlation test is shown in Figure 7, and the obtained correlation graph in Figure 8.

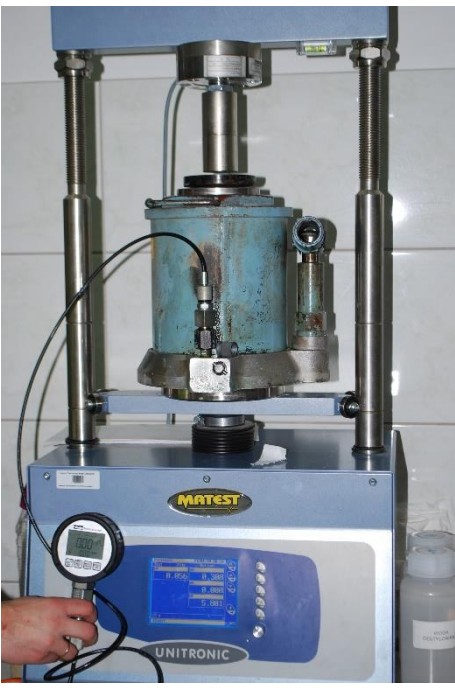

**Figure 7.** View of the correlation test of the manometer indications against the value of force exerted.

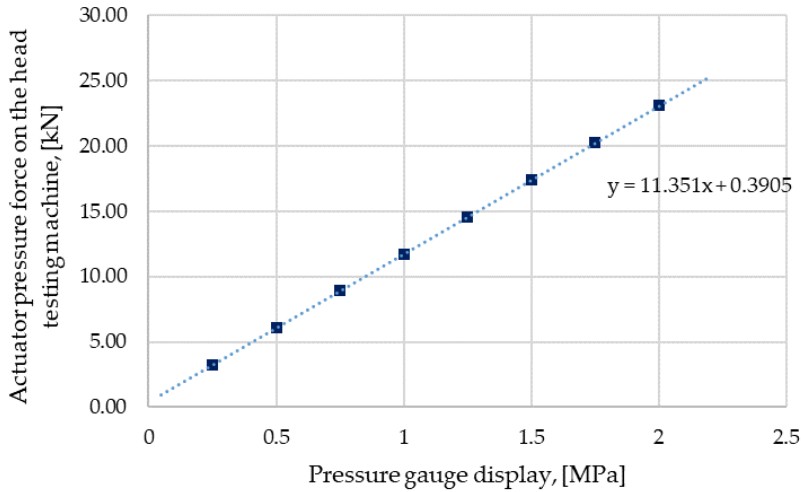

**Figure 8.** Correlation of pressure exerted by the actuator with the manometer indications.

The next steps of the procedure are as follows:

1.   Placing the wheel on the test bench,
2.   Bringing the pressure in the wheel to the set value,
3.   Placing the contrast on the wheel,
4.   Placing a clean screen between the wheel and the plate,
5.   Exerting wheel pressure on the plate with a given force.

Steps 2 to 5 were repeated for different values of wheel pressure and pressure values. Screens with a marked trace of the contact surface were placed in plastic shirts, to minimize the possibility of damage.

The tests negatives obtained were scanned at a resolution of 600 dpi in jpg format. Then, scans were attached in AutoCAD in the form of a trace. The trace was scaled to the actual dimensions, as a result of which, the unit of the length of the scanned wheel trace corresponded to the unit of the drawing length. The program created a contour of the scanned contact trace and its area was calculated using the functions available in the program. The outline was created several times, and the final result was the average of individual surface measurements. In this way, the impact of the error associated with the correct creation of the contour of the scanned contact trace was minimized.

## 3. Results

Measurements were made for four different types of tires. Tire pressure was determined on the basis of tire parameters and tire operating pressure. Measurements for the tire from the PZL M28 Skytruck/Bryza aircraft were carried out at a pressure in the wheel from 200 kPa to 600 kPa. For the tire from the Su-22 aircraft, the tire pressure range was between 200 kPa and 800 kPa. Tires from ASFT friction testers were tested with a pressure in the measuring wheel in the range from 210 kPa to 700 kPa. Detailed pressure values are presented in Table 1.

**Table 1.** Wheel pressure values during measurements for individual tire types (values given in kPa).

| PZL M28 Skytruck/Bryza | Su-22 | ASFTSmooth (ASTM) | ASFTSmooth-Tread (T520) |
|---|---|---|---|
| 200 | 200 | 210 | 210 |
| 400 | 400 | 280 | 280 |
| 500 | 600 | 350 | 350 |
| 600 | 800 | 420 | 420 |
| - | - | 490 | 490 |
| - | - | 560 | 560 |
| - | - | 630 | 630 |
| - | - | 700 | 700 |

Values of wheel pressure on the plate were determined based on the service load of individual wheel types. For tires from PZL M28 Skytruck/Bryza and Su-22 aircraft, the pressure values ranged from 3 kN to 23 kN. However, in the case of friction tester tires, these values ranged from 0.8 to 4 kN. Detailed values of the pressure are presented in Table 2.

**Table 2.** Values of wheel pressure on the plate during measurements for individual types of tires (values given in kN).

| PZL M28 Skytruck/Bryza | Su-22 | ASFT Smooth (ASTM) | ASFT Treaded (T520) |
|---|---|---|---|
| 3.23 | 3.23 | 0.8 | 0.8 |
| 6.07 | 6.07 | 1.4 | 1.4 |
| 8.90 | 8.90 | 2.0 | 2.0 |
| 11.74 | 11.74 | 2.5 | 2.5 |
| 14.58 | 14.58 | 3.0 | 3.0 |
| 17.42 | 17.42 | 4.0 | 4.0 |
| 20.25 | 20.25 | - | - |
| 23.09 | 23.09 | - | - |
| - | 25.93 | - | - |

As a result of measurements, 146 wheel and surface contact traces were obtained. Some combinations of pressure and pressure force could not be tested for safety reasons. The scanned negatives are presented in Figures 9–12.

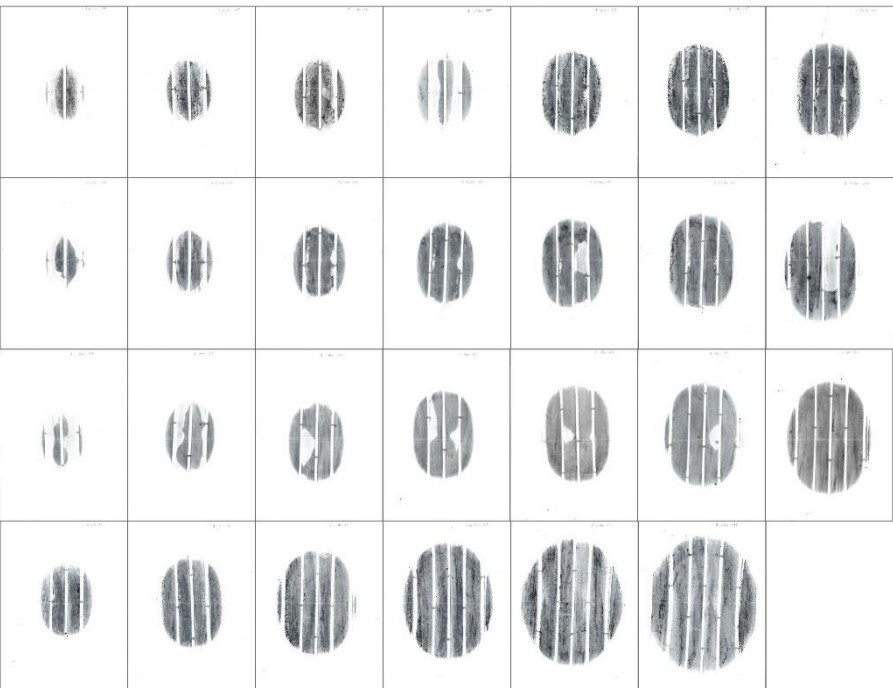

**Figure 9.** Negatives of wheel-to-surface contact areas for the PZL M28 Skytruck/Bryza aircraft wheel.

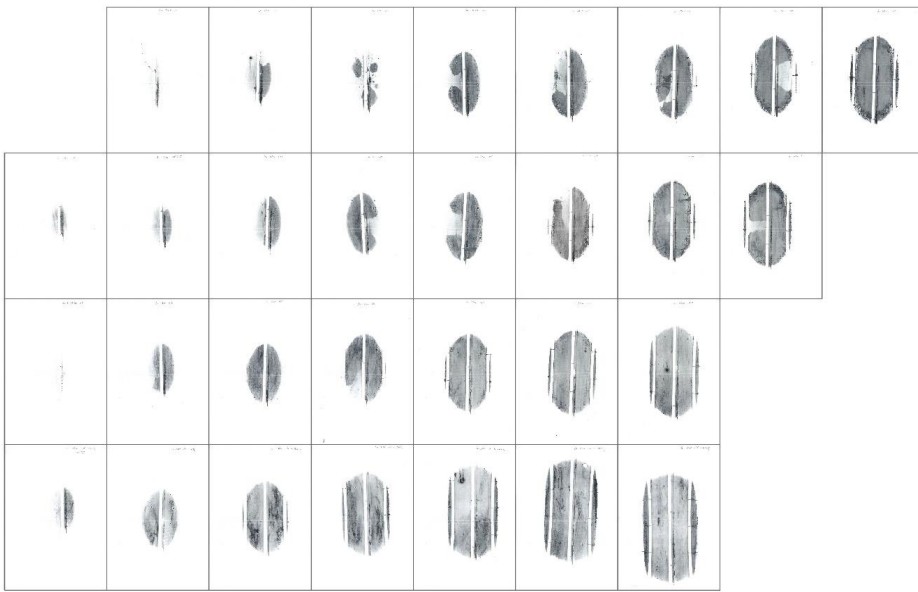

**Figure 10.** Negatives of wheel-to-surface contact areas for the Su-22 wheel.

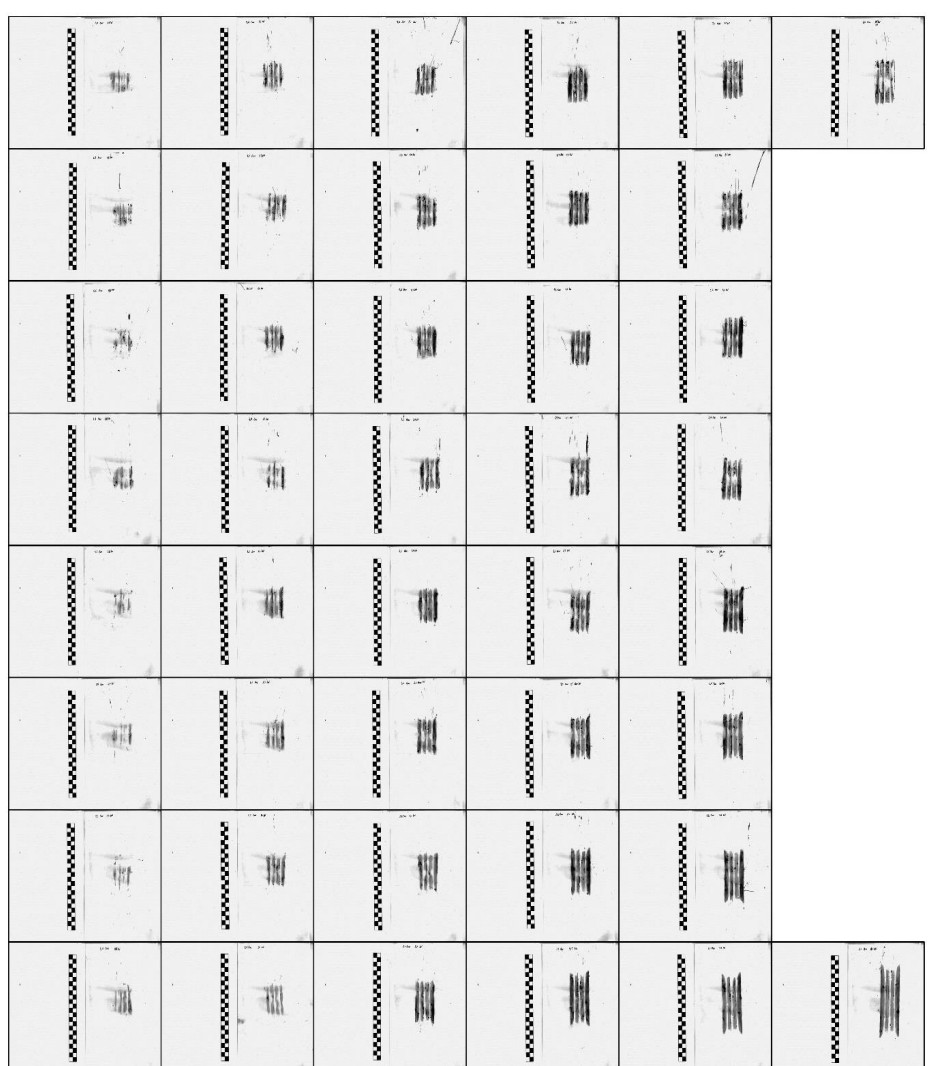

**Figure 11.** Negatives of wheel-to-surface contact areas for the ASFT tester wheel (T520 retreaded tire).

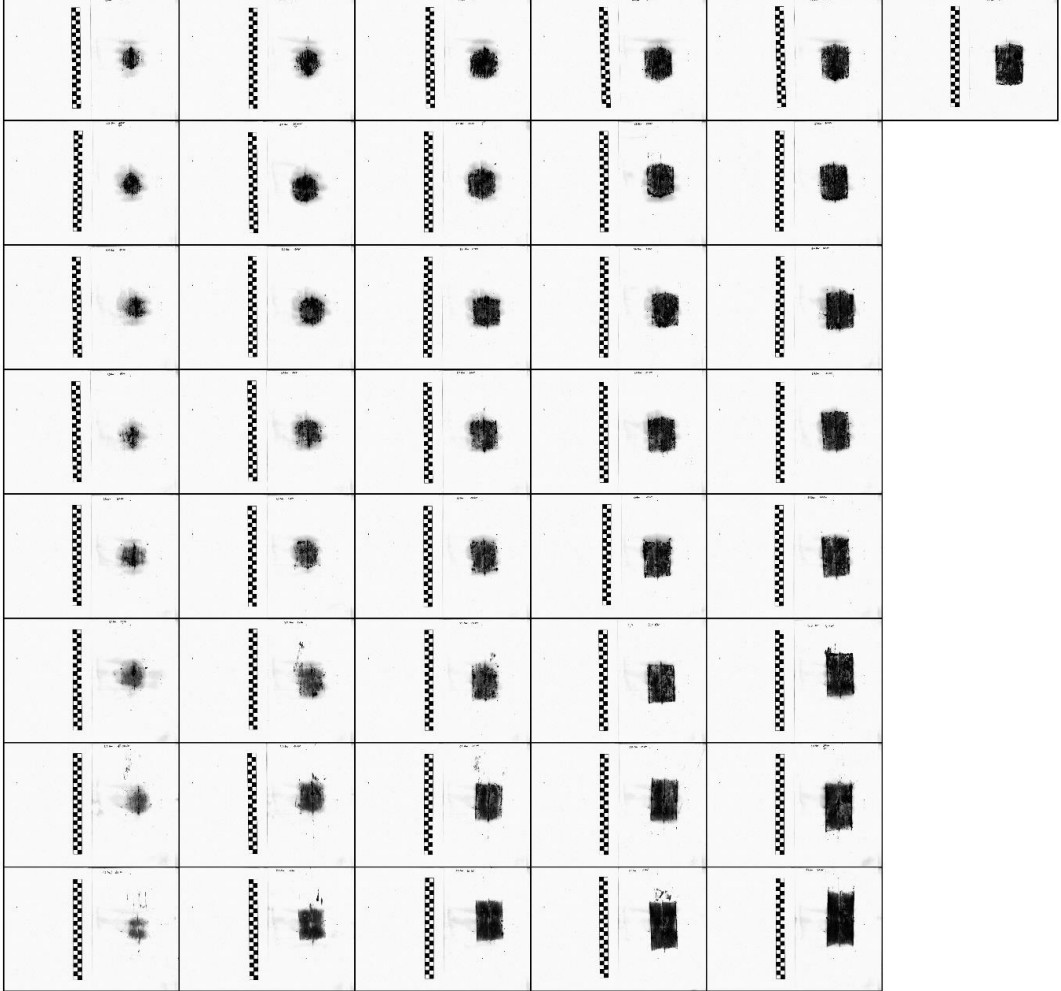

**Figure 12.** Negatives of wheel-to-surface contact areas for the ASFT tester wheel (ASTM smooth tire).

Each of the scanned negatives of the tire contact trace with the pavement was outlined in the AutoCAD envelope. Then, the surface area of the geometric figure thus formed was calculated. An example of an envelope is shown in Figure 13. The envelopes were created several times for the same circle and measurement conditions, and the results were averaged. The results for aircraft tires are collected in Tables 3 and 4, while ASFT testers tires—in Tables 5 and 6. On the contrary, Figures 14–17 show the relation between the aircraft wheels-surface contact surface and the pressure force. Each graph presents a different wheel pressure value. Each chart shows the results of tires testing from PZL M28 and Su-22 aircraft and the corresponding theoretical values calculated using Formulae (3) and (4). Values corresponding to those calculated from Formula (3) are marked on the graph as P/p (equation), while obtained from Formula (4) as [aircraft type] (equation). The wheel diameter of the PZL M28 plane is 72 cm, while the Su-22 plane—88 cm, and such values were taken for calculations.

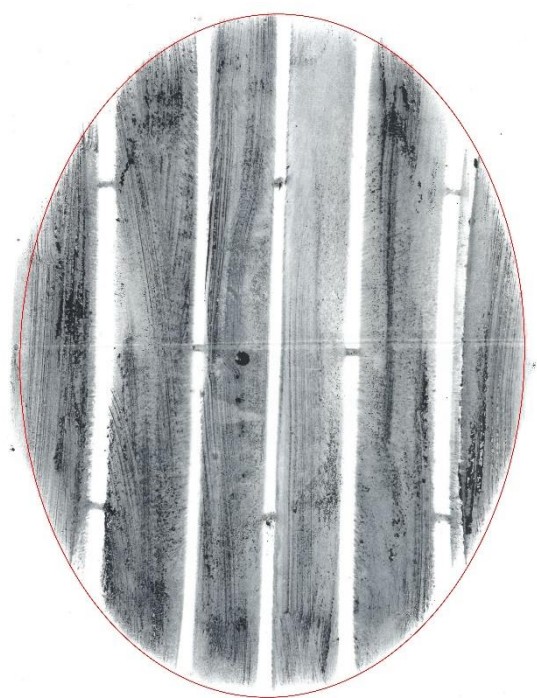

**Figure 13.** The envelope of one of the negatives (PZL M28 Skytruck/Bryza aircraft-pressure in the wheel 200 kPa, load 14.58 kN).

**Table 3.** Surface area values resulting from testing the PZL M28 Skytruck/Bryza aircraft tire (results in mm$^2$).

| Pressure<br>Force | | 200 | 400 | 600 | 800 |
|---|---|---|---|---|---|
| | | [kPa] | | | |
| 3.23 | | 9040 | 3856 | 5384 | 5039 |
| 6.07 | | 18,732 | 11,758 | 10,485 | 10,038 |
| 8.90 | | 27,790 | 16,908 | 14,773 | 14,503 |
| 11.74 | kN | 40,326 | 21,564 | 19,035 | 17,830 |
| 14.58 | | 46,753 | 27,358 | 22,844 | 20,438 |
| 17.42 | | 57,022 | 34,158 | 27,755 | 26,834 |
| 20.25 | | 67,608 | 39,574 | 33,554 | 29,352 |
| 23.09 | | - | 44,464 | 37,678 | 35,658 |

**Table 4.** Surface area values resulting from testing the Su-22 aircraft tire (results in mm$^2$).

| Pressure<br>Force | | 200 | 400 | 600 | 800 |
|---|---|---|---|---|---|
| | | [kPa] | | | |
| 3.23 | | 7270 | 3957 | 4075 | - |
| 6.07 | | 14,350 | 8669 | 5654 | 4296 |
| 8.90 | | 22,489 | 14,114 | 9550 | 9768 |
| 11.74 | | 28,848 | 19,849 | 13,412 | 12,483 |
| 14.58 | kN | 35,169 | 24,839 | 18,009 | 14,888 |
| 17.42 | | 41,506 | 29,134 | 22,710 | 18,077 |
| 20.25 | | 45,810 | 33,249 | 25,976 | 20,706 |
| 23.09 | | - | - | 28,889 | 25,129 |
| 25.93 | | - | - | - | 30,370 |

**Table 5.** Surface area values resulting from testing the ASFT tire-smooth tire ASTM (results in mm$^2$).

| Force Pressure | | 0.8 | 1.4 | 2 | 2.5 | 3 | 4 |
|---|---|---|---|---|---|---|---|
| | | kN | | | | | |
| 210 | | 2685 | 2938 | 4628 | 5561 | 6256 | 7371 |
| 280 | | 2069 | 3391 | 4084 | 4957 | 5674 | - |
| 350 | | 2293 | 3178 | 3975 | 4555 | 5058 | - |
| 420 | kPa | 2129 | 3201 | 3762 | 4420 | 4957 | - |
| 490 | | 2366 | 2816 | 3636 | 4306 | 4763 | - |
| 560 | | 2157 | 2974 | 3606 | 3923 | 4671 | - |
| 630 | | 2287 | 3262 | 3509 | 3949 | 4413 | - |
| 700 | | 2276 | 2918 | 3304 | 3930 | 4393 | 5037 |

**Table 6.** Surface area values resulting from testing the ASFT device-T520 retreaded tire (results in mm$^2$).

| Force Pressure | | 0.8 | 1.4 | 2 | 2.5 | 3 | 4 |
|---|---|---|---|---|---|---|---|
| | | kN | | | | | |
| 210 | | 3117 | 5108 | 6064 | 7241 | 8398 | - |
| 280 | | 3091 | 4845 | 5582 | 6466 | 7291 | - |
| 350 | | 3043 | 4510 | 5491 | 5890 | 6727 | - |
| 420 | kPa | 3029 | 4275 | 5045 | 5599 | 6184 | - |
| 490 | | 2686 | 4169 | 4923 | 5264 | 6130 | - |
| 560 | | 2552 | 4026 | 4750 | 5189 | 5728 | - |
| 630 | | 2359 | 3661 | 4360 | 5023 | 5555 | - |
| 700 | | 2066 | 3259 | 4354 | 4816 | 5383 | 6315 |

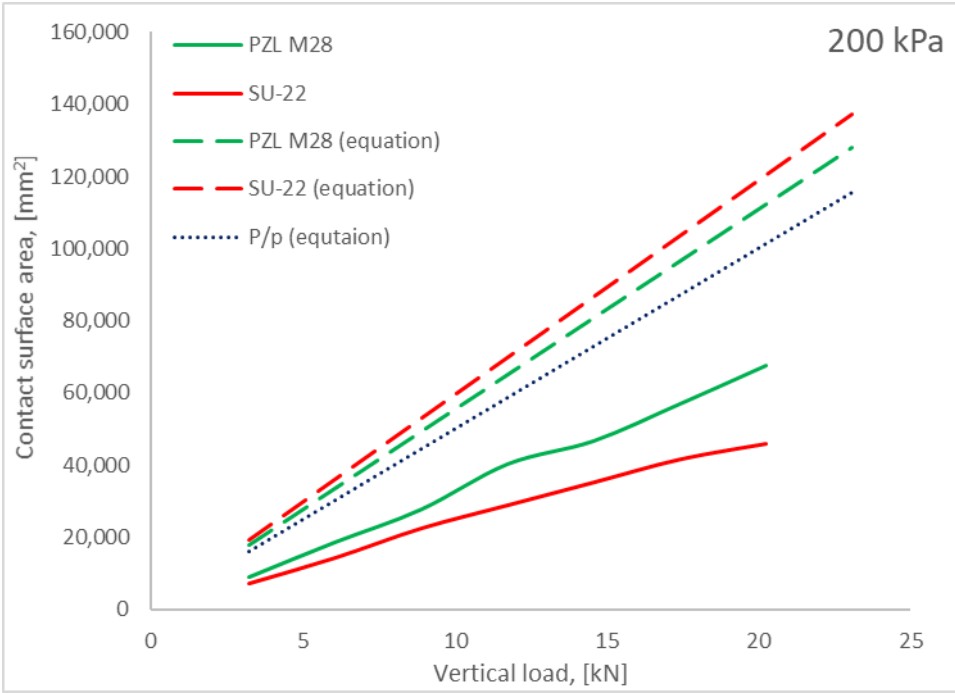

**Figure 14.** Contact surface area of the aircraft tire with the surface, in relation to the load for the wheel pressure of 200 kPa.

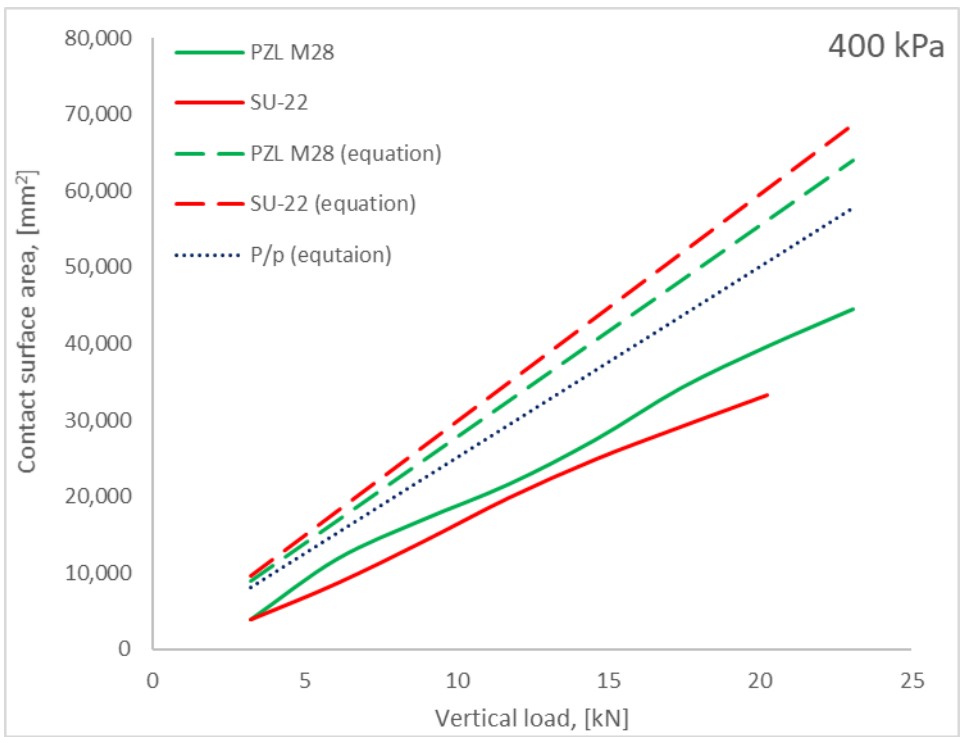

**Figure 15.** Contact surface area of the aircraft tire with the surface, in relation to the load for the wheel pressure of 400 kPa.

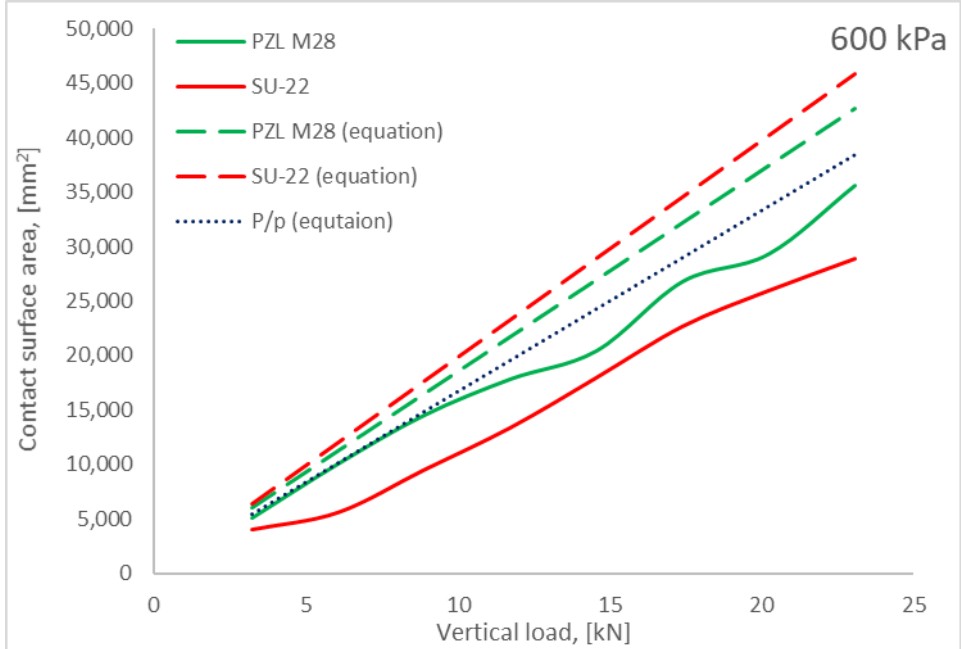

**Figure 16.** Contact surface area of the aircraft tire with the surface, in relation to the load for the wheel pressure of 600 kPa.

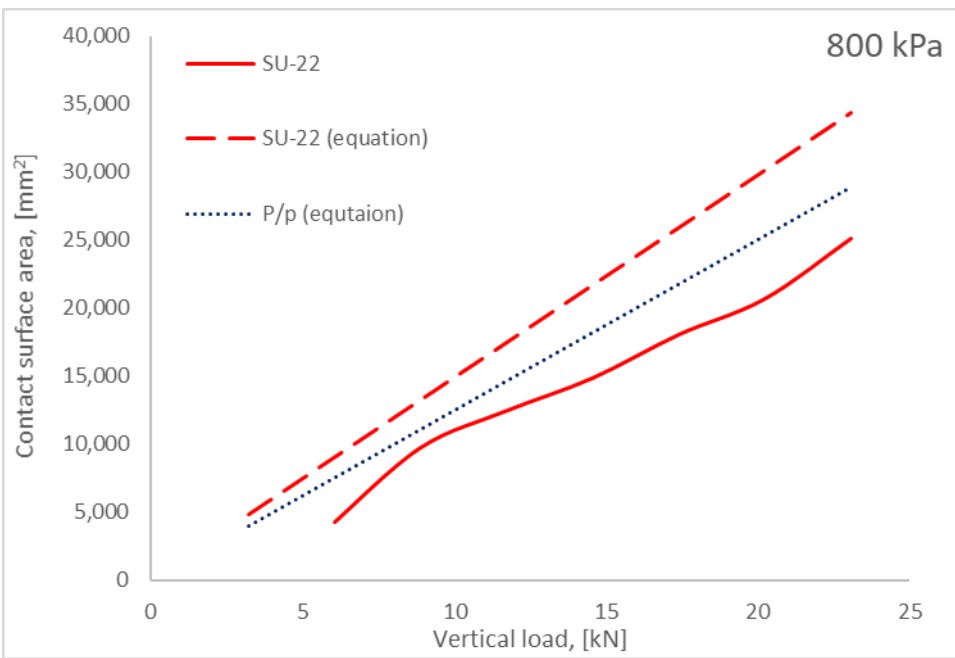

**Figure 17.** Contact surface area of the aircraft tire with the surface, in relation to the load for the wheel pressure of 800 kPa.

Figures 18–20 show the relation between the ASFT measuring wheels contact surface area and the load. Each graph corresponds to a specific, selected wheel pressure. In addition, each graph presents the theoretical values of the contact surface area resulting from Formulae (3) and (4). The graphs are designated as Equations (1) and (2) respectively.

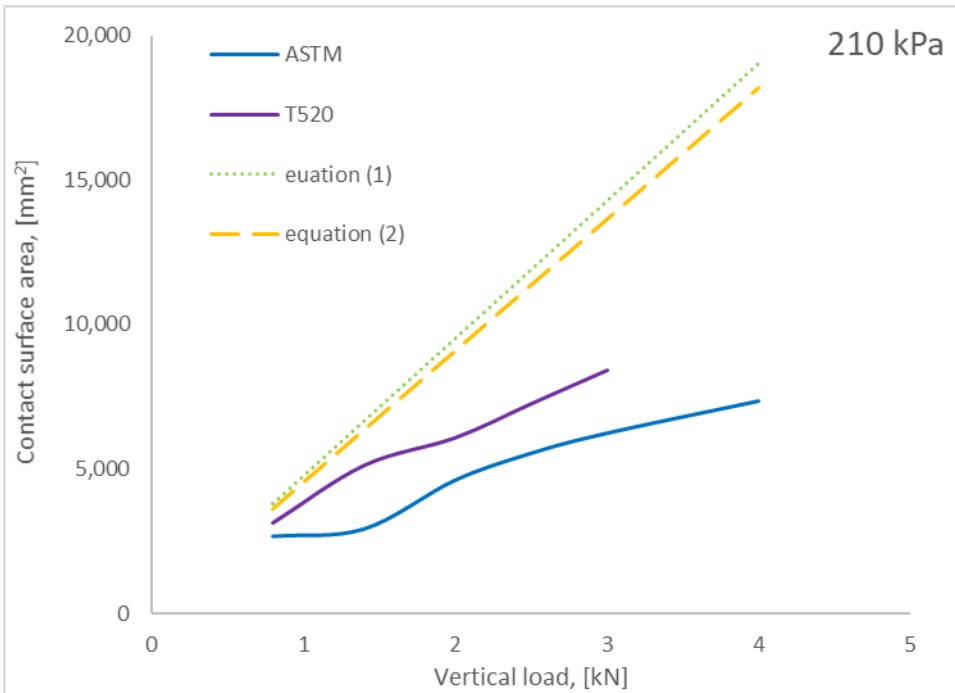

**Figure 18.** Contact surface area of the ASFT tires with surface, in relation to the load for wheel pressure of 210 kPa.

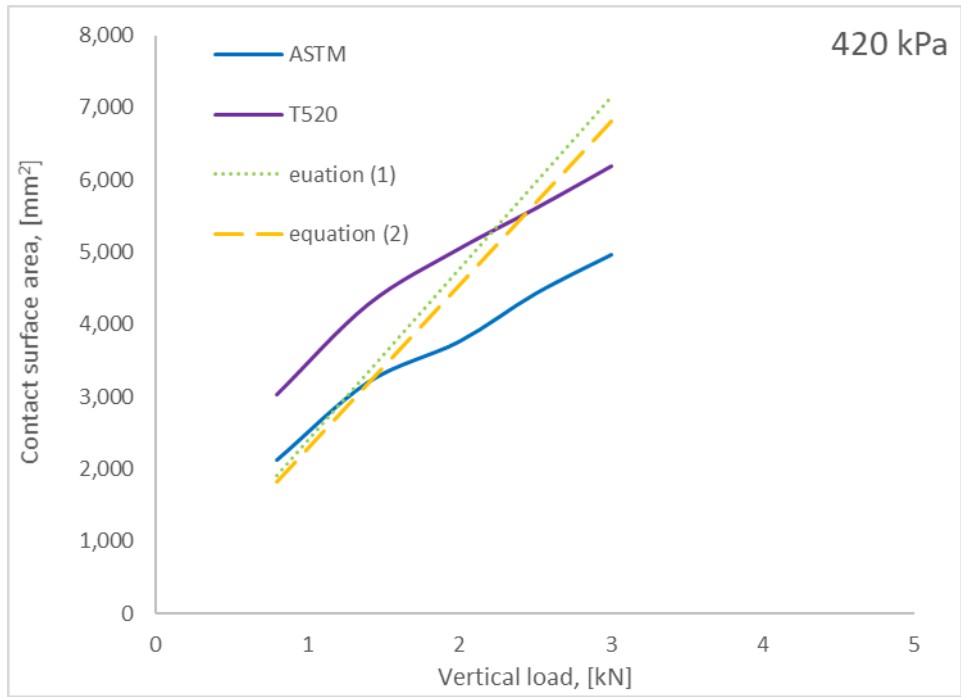

**Figure 19.** Contact surface area of the ASFT tires with surface, in relation to the load for wheel pressure of 420 kPa.

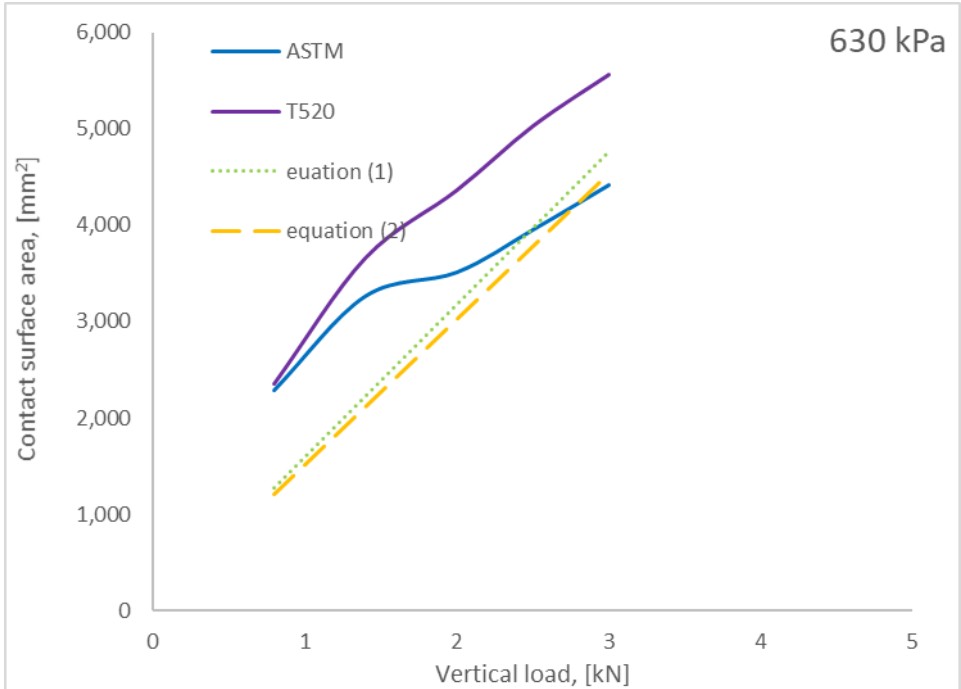

**Figure 20.** Contact surface area of the ASFT tires with surface, in relation to the load for wheel pressure of 630 kPa.

## 4. Discussion

Tire contact surface area tests have been conducted for many years in various contexts. Researchers analysed the impact of tire pressure on the stress formation in the pavement; the car tire contact surface area was studied in relation to the load [1–3]. The contact surface area was also examined in terms of the anti-skid properties of the pavement [4–7]. Stress distribution in the soil at the wheel-contact

surface was also analysed [14–17]. Such a varied approach to the phenomenon presented in the article shows how important the contact of the tire with the road surface is.

The authors of this work presented the results of a study of the relation between the aircraft tires' contact area with the road surface, and the pressure force of the wheel on the surface and the pressure in the wheel. The study included two tires—one from the PZL M28 Skytruck/Bryza aircraft, the other from the Su-22. The tires from the ASFT airport friction tester were also tested—the T520 retreaded tire and the ASTM smooth tire.

Analysing the results presented above, it should be noted that in no case did the results of the wheel-surface contact area obtained during the test coincide with the theoretical values derived from the Formulae (3) and (4). As shown in Figures 14–17, aircraft tires obtained lower than theoretical values in every configuration. Moreover, theoretically, the contact surface for a Su-22 tire should have a larger surface area than the PZL M28 tire. In practice, the study revealed a completely opposite relationship. In each case, the contact area of the PZL M28 aircraft tire had a higher value than for the Su-22 aircraft tire. In addition, as the pressure in the measuring wheel increased, the differences between the surface area between individual types of tires and between their theoretical values decreased.

It should be noted that, in each case, the wheel contact surface with the surface increased with increasing pressure. The dependence of the surface area on the wheel pressure was the opposite. As the pressure in the wheel increased, the surface area decreased.

Based on the test results presented, it can be argued that the contact surface of the tire with the surface is influenced by factors other than just the vertical load and pressure in the wheel. In addition, the relationship resulting from Formula (4), taking into account the diameter of the tire, is not consistent with that obtained in the tests. This allows the theory to be drawn that, apart from vertical load, wheel pressure and tire diameter, there are other factors determining the size of the plane's contact surface with the surface. A surface plot of research results (Figure 21) presents the relationship between the contact surface and load/pressure values.

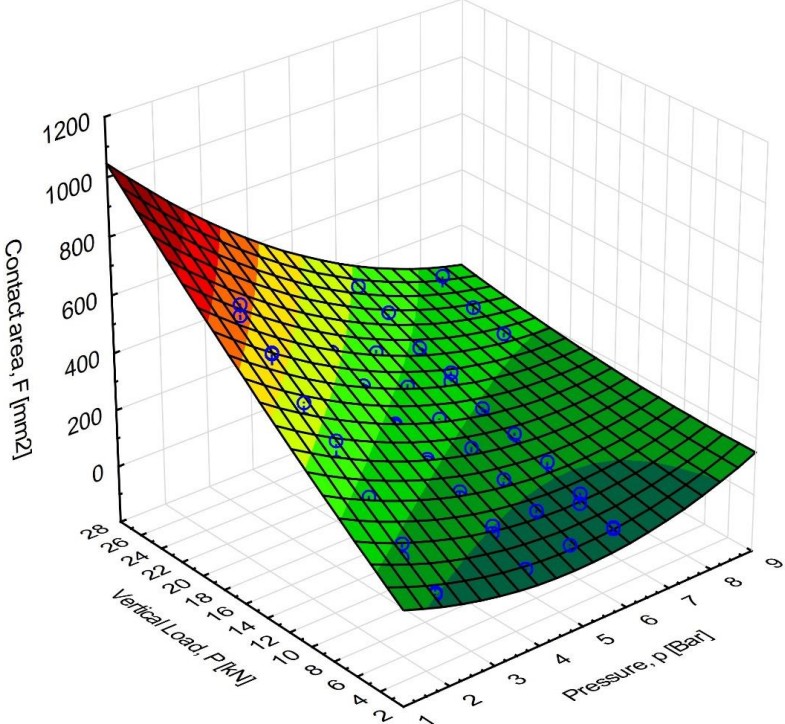

**Figure 21.** Contact surface area for aircraft tires.

For presented research results, the approximate model of the tire-pavement contact surface area looks like (5):

$$F = 130 - 70p + 34P + 8p^2 - 3,9pP + 0,16P^2 \, , \tag{5}$$

where:

F—tire-surface contact area [mm$^2$],
P—vertical load of the wheel on the surface [kN],
p—wheel pressure [Bar].

Based on research results and the theoretical base (3) and (4), the model of the tire-pavement contact area may look like (6):

$$F = d_T \frac{P}{p} + c_T \, , \tag{6}$$

where:

F—tire-surface contact area [m$^2$],
P—vertical load of the wheel on the surface [N],
p—wheel pressure [Pa],
$d_T$, $c_T$—tire shape indicators.

Furthermore, in the case of friction tester tires, the relationship between vertical load, wheel pressure and contact surface is not clear. Both tested tires have the same dimensions, and according to Formula (4), they should present the same contact area values for the same measurement conditions. In fact, the T520 tire generates a smaller contact area than the ASTM tire. In addition, at a wheel pressure of 350–420 kPa (Tables 5 and 6), the values obtained in the study are close to the theoretical values derived from the Formulae (3) and (4), as shown in Figure 19. At a lower pressure (Figure 18), the contact surface obtained is lower than the theoretical, while at a higher pressure (Figure 20), the contact surface exceeds the theoretical values.

## 5. Conclusions

The wheel contact surface with the airport surface is important for the safety of flight operations in the ground manoeuvring area. It also significantly affects the magnitude of stresses arising in the structure, which affects the durability and reliability of the airport pavement. Contact surface tests can be found in many references, some of which are cited in this article. Research was conducted in many different contexts. Among others, the impact of various factors on the contact surface, its shape, and the manner of stress distribution were analysed, but parameters such as the anti-skid properties of the pavement were also interested.

In this study, the authors showed that commonly used formulae for the theoretical contact surface differed from the results obtained during laboratory tests, in consideration of the impact of tire pressure and pressure force between the wheel and the airport pavement on the contact surface area. It has been shown that wheel pressure, pressure force and tire size are not the only parameters affecting the wheel's contact surface area.

The research results presented in the article concern a part of a large research project aimed at developing the device and methods for the continuous measurement of the load capacity of natural airport pavements. The tests were carried out on a measuring stand created for this purpose, and the results are the basis for the selection of the constructed device's relevant elements.

In the future, the authors plan to analyse other types of tires from different types of aircraft. Thus, a more extensive database will be created, that will allow a more detailed analysis of the impact of various parameters on the shape and contact surface of the wheel with the road surface. It is estimated that the steps taken will allow the development of a theoretical model that will easily, with relatively high accuracy, estimate the actual contact surface area of the aircraft wheel with the airport pavement.

**Author Contributions:** M.W.: Conceptualization, Writing—Review and Editing; K.B.: Methodology, Writing—Review and Editing; P.P.: Resources, Data curation, Writing—Review and Editing; P.I.: Formal analysis, Writing—Original Draft and Editing. All authors have read and agreed to the published version of the manuscript.

**Funding:** This research was funded by NCBiR under the project no POIR.01.01.01-00-0239/19 "Unmanned, autonomous platform for continuous measurement of natural airfield pavement's load bearing capacity".

**Conflicts of Interest:** The authors declare no conflict of interest.

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
