# Peer review of "Analysis of the Actual Contact Surface of Selected Aircraft Tires with the Airport Pavement as a Function of Pressure and Vertical Load"

_coatings, doi:10.3390/coatings10060591_

Round 1

Reviewer 1 Report

The study has only analyzed the trend of contact. I think more theoretical analysis should be given. The relationships between contact and other factors should be approximated at least.  

Author Response

Thank you for revision.

The article was improved:

  • There are little changes in abstract,
  • Specific information about tires was add,
  • Information about temperature during tests was add in line 195,
  • Additional results analys was add,
  • Additional references in "Discussion" were add.

This is the first stage of research and based on obtained results there is not a good idea to approximate precise model of contact area. However general model was add to discussion. Broader research is underway.

Reviewer 2 Report

The authors have for sure done a lot of work to analyze the contact between a aircraft tires and airport pavement. However, I don't think that the manuscript in the present form can be accepted for publication.

  1. It is not clear for me why the authors assume that Coatings is the right journal to publish their work. As the name of the journal unambiguously indicates, papers published in Coatings should have a focus on coatings. This is clearly not the case for this work. Another clear indication that Coatings is not a suitable journal for this manuscript is that authors did not cite a single paper published in Coatings or any other journal focusing on surface modification or coating technologies. This manuscript is clearly out of the scope of the journal Coatings. Authors are advised to submit the work to a journal which is more suitable for such topics.
  2. The manuscript reads more like a technical report than like a scientific paper. Although the authors presented a nice overview on the state-of-the-art in the introduction, there is hardly any interpretation of the results. In many cases, there is not even a description of the results presented in the figures and tables!
  3. The tires tested are not described well. Just mentioning that these are the tires of certain aircrafts does not have any significant and valuable meaning for the readers, where they can learn something. There is not even an attempt to generalize the findings, to increase the value for readers interested in the behavior of tires of other aircrafts.
  4. There is no discussion of the results obtained, taking into account the existing literature. In the section entitled Discussion, authors did not use a single reference! Why did they spend a lot of time to present the nice overview in the introduction, when the cited references are not used again? The authors completely fail in comparing their findings to others and to inform about the implications of their work.

Author Response

(The authors gave the same response as above.)

Reviewer 3 Report

The manuscript entitled “Analysis of the actual contact surface of selected aircraft tires with the airport pavement as a function of pressure and vertical load” focused on the characterization of the surface area between the tires and the pavement surface. This reviewer believes that the topic is interesting, and the manuscript is worth publishing. Please consider addressing the following comments to improve the quality of this research work.

  • The abstract needs to be significantly revised. The abstract should include the research gap, the importance of the issue being introduced, method(s) applied in the research to close the gap, the outputs of the research. This abstract lacks many parts of a scientific one.
  • Page 2, line 44-45: “The finite element method (FEM) was also used for the analysis of the research results. The analysis results confirmed that for a given pressure, the contact surface decreased with the increase of the tire pressure “. Reference is needed.
  • Figures 2 to 8 were good choices for this manuscript.
  • The methodology is well described in section 2.
  • Are all the measurements conducted at the same temperature? Also, please mention the temperature.
  • Do you think that the actual pavement deformation will affect the measurements? If so, what would be the effects(s)?

Editorial Comments:

  • “In [4] the research aimed at correlate the tire-road contact”-Rewrite.

Author Response

Thank you for revision.

The article was improved:

  • There are little changes in abstract,
  • Specific information about tires was add,
  • Information about temperature during tests was add in line 195,
  • Additional results analys was add,
  • Additional references in "Discussion" were add.

The statement in lines 44-45 refers to [2] position in bibliography. Word "also" can be misleading so it was deleted.

Pavement deformation could affect the measurement and it could increase the contact area. It has great meaning when we consider natural pavements such as soil or grass pavement.

Reviewer 4 Report

This is a very interesting manuscript that addresses important issues related to the relation between the aircraft tires contact area with the road surface, and the pressure force of the wheel on the surface and the pressure in the wheel.

The paper is well structured and clearly written.

Although the English language is not bad, a revision would be advisable because the manuscript is difficult to follow.

Check the quality of the figures!

From these reasons, I think that this manuscript, after a minor revision, could be accepted to be published by the "Coatings” Journal.

Author Response

(The authors gave the same response as above.)

Round 2

Reviewer 1 Report

In my opinion, I think the paper should use simulation to compare the result of the experiment. Moreover, the highlights of this paper should be discussed. Compared with other methods. How about the advantages of yours.

Author Response

The reviewed article is a piece of a large scientific and research project aimed at building an autonomous platform for continuous assessment of load bearing capacity of natural airport pavements. Presented in the article results are preliminary tests that are the basis for choosing the proper measuring wheel for the above mentioned device. One of the criteria for wheel selection was the actual contact area with the road surface. Due to the lack of literature regarding wheel actual contact area with the airport pavement and discrepancies with theoretical values ​​obtained as a result of calculations using known mathematical models, the authors attempted tests in laboratory conditions. In this case, a rigid airport pavement was simulated using a non-deformable steel plate. The obtained results were compared with theoretical values, and the relevant commentary was included in the article. As part of the project, further tests of the selected measuring wheel are planned in field conditions and to determine the actual contact area of the wheel with the deformable surface. The cited literature does not contain research results in the analyzed area, i.e. for aircraft shin wheels.

Presented test method is not the core of the article. Authors focued on obtained results and the main part of the article is correlation between tire pressure, load and contact surface area. The authors' intention was to perform the tests in controlled laboratory conditions, therefore their own measuring station was built.

Reviewer 2 Report

I do not think that the authors significantly improved their manuscript. They did not even reply to my major points of criticism. I still think that manuscripts to be published in the journal "Coatings" should somehow deal with the topic "coatings". This is not the case here, the authors did not try to present a single reason why this work should be published in this particular journal. I also think that the manuscript is rather a technical report than a scientific paper. The authors did not even try to compare their results with those available in the literature. Technical reports like this one should not be published in a peer-reviewed scientific journal. 

Author Response

The reviewed article is a piece of a large scientific and research project aimed at building an autonomous platform for continuous assessment of load bearing capacity of natural airport pavements. Presented in the article results are preliminary tests that are the basis for choosing the proper measuring wheel for the above mentioned device. One of the criteria for wheel selection was the actual contact area with the road surface. Due to the lack of literature regarding wheel actual contact area with the airport pavement and discrepancies with theoretical values ​​obtained as a result of calculations using known mathematical models, the authors attempted tests in laboratory conditions. In this case, a rigid airport pavement was simulated using a non-deformable steel plate. The obtained results were compared with theoretical values, and the relevant commentary was included in the article. As part of the project, further tests of the selected measuring wheel are planned in field conditions and to determine the actual contact area of the wheel with the deformable surface. The cited literature does not contain research results in the analyzed area, i.e. for aircraft shin wheels.

I would like to point out that the article was qualified for a special issue of "Pavement Surface Coatings" and according to information on the journal’s website, its scope covers the issues raised. The research work concerns the problem of diagnostics of natural airport pavements in the context of safety management and operation process, and the article constitutes its essential part. Obtained test results, through the shown discrepancies between theoretical and real relationships, can also outline new trends in the artificial pavements design, taking into account the special aspect of flight safety.

Round 3

Reviewer 1 Report

It can be accepted.

Reviewer 2 Report

Thank you for providing the information that this manuscript is intended to be included in a special issue on pavement coatings. It would have been nice if this information would have been provided by the publisher or at least by the authors after the first review round. This could have saved time in reviewing and publishing this manuscript.

With this missing information, I recommend to publish the manuscript in Coatings.